

# Mars sub-millimeter sensor on micro-satellite: sensor feasibility study

Richard Larsson[1], Yasuko Kasai[1], Takeshi Kuroda[1], Shigeru Sato[1], Hiroyuki Maezawa[2], Yutaka Hasegawa[3], Toshiyuki Nishibori[4], and Shinichi Nakasuka[5]

[1]National Institute of Information and Communications Technology, Tokyo, Japan
[2]Osaka Prefecture University, Osaka, Japan
[3]Institute of Space and Astronautical Science, Japanese Aerospace Exploration Agency, Tokyo, Japan
[4]Research and Development Directorate, Japanese Aerospace Exploration Agency, Tokyo, Japan
[5]Tokyo University, Tokyo, Japan

*Correspondence to:* Richard Larsson (ric.larsson@gmail.com)

**Abstract.** We are planning a mission to Mars using a micro-satellite to carry a sub-millimeter sensor to orbit and have performed a feasibility study to determine what we can measure and to what success. The sensor will measure atmospheric molecular oxygen, water, ozone, and hydrogen peroxide to retrieve their volume mixing ratios and change over time. The sensor will also retrieve the temperature field, the wind field, and the magnetic field at various levels of success. The expected
5   measurement errors for molecular oxygen is below 100 ppmv in limb view below 50 km, with 20 ppmv for near surface measurements. For water in limb-view, the retrieval errors are below 1 ppmv with a detection limit of a few tens of ppbv. For ozone the limits are at 2 ppbv, and for hydrogen peroxide the retrieval limits are in the range of 1 ppbv. In nadir-viewing geometry, the expected errors in the column are worse but not by much since we can keep integrating the signal from the same area for a long time, though the vertical resolution clearly suffers.

## 1   Introduction

We are in the works to build and fly a sub-millimeter sensor to Mars using a micro-satellite platform. The goal is to launch in 2020 or 2022, and the main scientific target is to measure changes in Martian molecular oxygen over time. The sensor is based on a previous proposal of a more advanced version of the instrument by Kasai et al. (2012), and our working name for the sensor is the Far-InfraRed Experiment Minimized (FIRE-mini). As Kasai et al. (2012) suggests, the advantages of sub-
15   millimeter technology for Martian remote sensing is that the radiation at sub-millimeter frequencies are mostly unaffected by atmospheric dust content, and that the radiation observed is passively emitted by the target atmospheric gases. The sensor will be able to measure molecular oxygen, water, hydrogen peroxide, ozone, the temperature field, the wind field, and the strongest crustal magnetic fields. Even inside dust storms and during nighttime.

Martian molecular oxygen have been measured several times by missions like Viking, Herschel, Curiosity, and MAVEN.
20   Carleton and Traub (1972) measured a global molecular oxygen profile of 1300 ppmv, Hartogh et al. (2010) measured a constant 1400 ppmv profile in whole disk measurements but remarked that there could be a higher concentration near the surface,



Mahaffy et al. (2013) measured a constant profile of 1450 ppmv in ground-based measurements, and Sandel et al. (2015) measured up to 4000 ppmv at altitudes of 90 to 120 km in limb measurements at nanometer wavelengths. The discrepancies are small between most of these but important and should be studied in more details. Molecular oxygen acts as the chemical background that oxidize water and various hydrogen radicals. Low altitude variations have not been measured in detail over

time. The measurements by Sandel et al. (2015), clearly shows that the idea of a constant profile does not work at altitudes above 90 km in stellar occultations. Our sensor will not be able to confirm these measurements because the altitude range is above our reach. We will be able to see if the variations observed at 90 km are reflected near the surface, and limit the altitude range at which the relative oxygen concentration starts increasing.

This paper is dedicated to show the feasibility study we performed to test the sensor design of FIRE-mini. There will be other

papers about the mission to describe the details of the orbit insertion and retention, and to give a broader scientific overview of the mission as a whole. This paper shows the forward simulations of the expected observations, and the error estimations from a simple retrieval setup. The next section describes the method of how we set up our simulations — it also discusses some of the limitations that we have encountered. After the description of our method, we show our results, discuss their consequences, and give our concluding remarks.

## 2 Method

This section goes through the basic assumptions we made to perform the simulations. This includes orbit considerations, sensor design, spectroscopic modeling, and retrieval procedure.

### 2.1 Orbit

The sub-millimeter sensor will be carried to Mars on a satellite weighing less than 100 kg. Such a small satellite needs special

means to enter orbit. The selected method to achieve orbit is to use the atmospheric drag to perform aerocapture. Aerocapture has never been attempted successfully before — as far as we are aware — so final orbital parameters are to some extent uncertain. A future work will discuss the orbit insertion and retention in details. For this work, we have opted to work simply with two sets of observations taken from an orbit with the Kepler elements semi-major axis of $6150\,\mathrm{km}$ and eccentricity of 0.5. This gives a periareion altitude of $400\,\mathrm{km}$ and an orbit time of 5 hours and 20 minutes. We will show simulated observations

as well as estimated errors for retrieved atmospheric quantities in limb-view geometry near the periareion and in nadir-view geometry near the apoareion. The final orbit could differ from the one above, but the feasibility of the sensor is not strictly dependent on the details of the orbit. As long as the periareion is not too high, it will merely affect our spatial resolution.

### 2.2 Sensor

A schematic sketch of the instrument can be seen in Figure 1. The spectrometer we plan to use is the chirp transform spectrom-

eter designed for the Jupiter icy moons explorer's sub-millimeter wave instrument, with 10000 channels over a 1-GHz range (for examples on this type of spectrometer, see, e.g., Hartogh, 1997). The local oscillator will be at 481.15 GHz with a central





intermediate frequency of 6 GHz. There will be no suppression of either of the sidebands, so the measured radiation will be from between 474.65-475.65 GHz and 486.65-487.65 GHz. The system noise temperature is expected to be about 2000 K in double-sideband mode. The antenna is planned to be a 30 cm large and made of carbon fiber reinforced plastic. This achieves a 10 km vertical footprint at altitudes below about 400 km, with a resolution half-width of about $0.14°$.

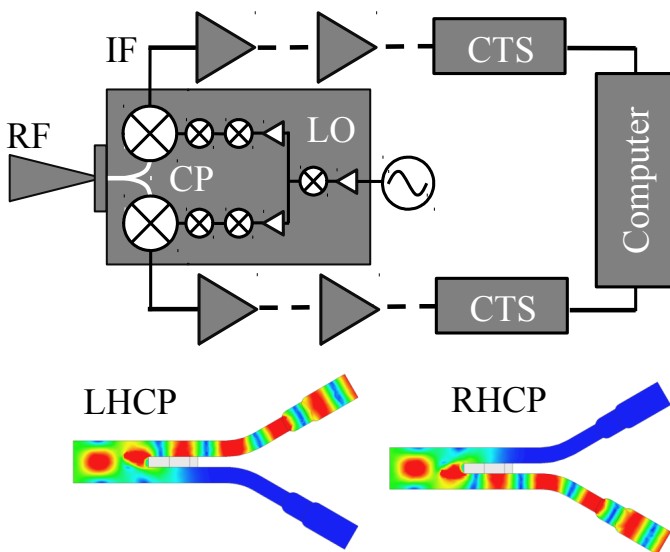

**Figure 1.** Sketch of sensor electronics design. The radio frequencies (RF) between 474.65-475.65 GHz and 486.65-487.65 GHz enters from the left in the figure and is split by a circular polarizer (CP) into left-handed, and right-handed, circular polarization (LHCP, RHCP) as demonstrated in the colored plots above. The signal is then mixed with the signal of a local oscillator at 481.15 GHz, to lower the frequency to a measurable range. At an intermediate frequency of 6 GHz the chirp transform spectrometer turns this analog signal into a digital signal that is fed into a computer for preparations and to be sent back to Earth.

5   We plan to use two identical receiver systems with the same setups but fed different states of polarization. The polarization states will be separated using a feed horn antenna. This way, if one of the receiver chains experience technical issues, we can use the other to keep measuring the atmosphere. In addition, while both chains work we can measure the magnetic field. Molecular oxygen is affected by the Zeeman effect (Zeeman, 1897), so it is possible to sense the strongest crustal magnetic field through the state of polarization (for more details, please see Larsson et al., 2013, 2014, 2017).

10   Because of the dual receiver systems, if we take a single measurement every second, the measurements alone produces about 160 kbps of data before any compressions are applied. This exceeds our maximum transfer rate, so we will have to perform limited data reduction on-board the spacecraft. The software for this is not ready yet. The only reduction considered in this work is an averaging by pairing immediate neighboring channels to increase the effective width of a channel to 200 kHz. This is still well below the line width of the absorption lines that we consider here, so no physics is lost.



## 2.3 Forward Model

We use the Atmospheric Radiative Transfer Simulator (ARTS; Buehler et al., 2005; Eriksson et al., 2011) for all forward simulations. ARTS is a fully three-dimensional model with full polarization capabilities that have been used in numerous studies. Please see the two cited articles, other articles citing them, and the source code — available via a copyleft license at

5    `www.radiativetransfer.org` — to understand the radiative transfer method of ARTS in more details. All of the data used in this study can be found via aforementioned link.

    The atmosphere is using temperatures and pressures based on Laboratoire de Météorologie Dynamique's global circulation model (Forget et al., 1999). The temperature profile is shown in Figure 2. The magnetic field is from Cain et al. (2003). Gases are simplified from the circulation model's profiles, and are kept at constant volume mixing ratio in each simulation. The

10   volume mixing ratios of the gases are at $95.32\,\%$ of carbon dioxide, 1400 ppmv of molecular oxygen, 50 ppmv of water, 10 ppbv of ozone, and 10 ppbv of hydrogen peroxide. Other atmospheric species expected on Mars cannot be seen in our spectral range. Using constant atmospheric mixing ratio profiles is by no means a good representation of the atmosphere, but is sufficient to estimate how well the gases can be retrieved at different altitudes. We expect there to be a large variations in the trace gases (i.e., water, ozone and hydrogen peroxide), so our setup must be responsive to this. We also expect there to be some

15   variations in the molecular oxygen profile. In order to emphasize detection limits, we perform the simulations at a factor 100 times less of the trace gases.

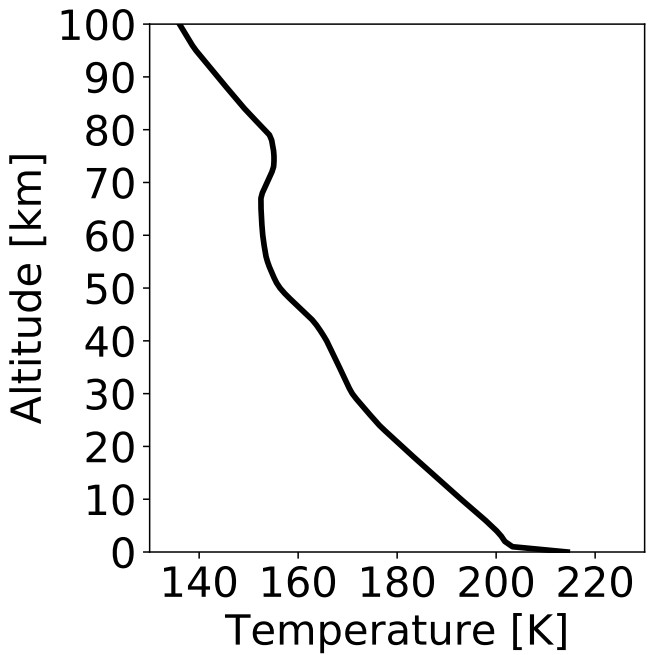

**Figure 2.** Temperature profile used in the simulations.



A spectroscopic suite suitable for Mars was developed by Buehler et al. (2017), which we use for our simulations. It computes pressure broadening and shifting per atmospheric species rather than from a predefined atmospheric composition. We also use the carbon dioxide collision-induced absorption from Gruszka and Borysow (1997) as a continua shown in Figure 3. This continua is key to low altitude limb views, and is the main reason we are focusing on the $400\,\mathrm{GHz}$ range rather than higher frequencies. Note that the continua is effectively twice as strong as normal since we will not suppress the lower or upper sidebands. A big problem with this continua is that it is only defined down to a temperature of $200\,\mathrm{K}$. For lower temperatures we simply extrapolate to these using the Gruszka and Borysow (1997) code by ignoring the warnings. Despite these issues, we still choose to include the continua since ignoring a potential 20-120 K signal would be catastrophic for the science of the mission. Clearly, the carbon dioxide continua has to be studied more for Mars atmospheric conditions since its influence is notably strong. Even if our extrapolation is the cause of most of the absorption that we see (because of some model artifact at lower temperatures), it is necessary to confirm that this is the case, and to limit the influence of the continua.

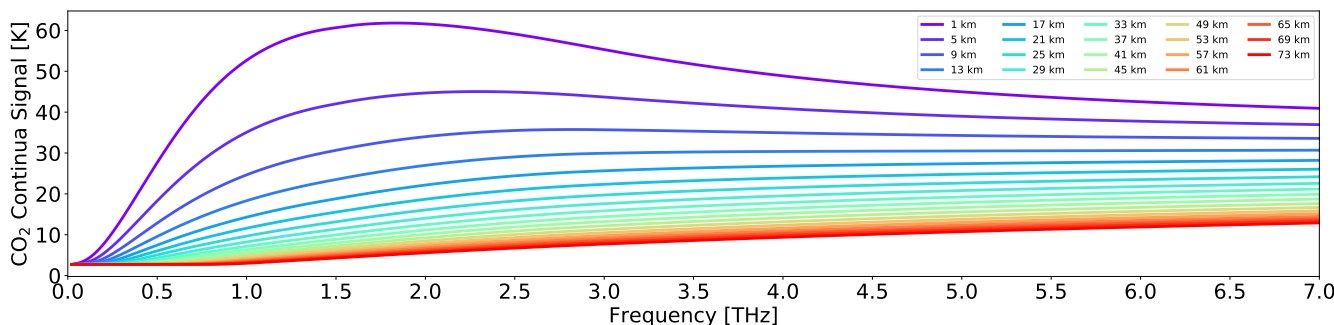

**Figure 3.** Limb pencil view of Mars considering only the carbon dioxide continua. Pencil tangent altitudes are indicated by the legend. The increase in the signal at higher altitudes and higher frequency is probably a consequence of the extrapolation of the model parameters.

## 2.4 Error Analysis

We use the optimal estimation method described by Rodgers (2000) to predict at what level of certainty we can measure atmospheric parameters. This is done from assuming a linear error. Errors tend to be linear near the target even if the retrieval process or underlying physics is non-linear. This error analysis is from

$$\mathbf{e}_x = \mathbf{S}_a \mathbf{J}^\top \left( \mathbf{J} \mathbf{S}_a \mathbf{J}^\top + \mathbf{S}_e \right)^{-1} \mathbf{e}_y, \qquad (1)$$

where $\mathbf{S}_a$ is a simplified a priori covariance matrix assuming independence between the variables, $\mathbf{J}$ is the Jacobian matrix, $\mathbf{S}_e$ is a diagonal error covariance matrix, and $\mathbf{e}_y$ is a Gaussian realization of a measurement error. This realization is repeated a number of times to estimate the error on retrieved parameters. The a priori covariance matrix is simplified to be block-diagonal, set to have a $10\,\mathrm{km}$ exponentially decreasing correlation, with estimated diagonal elements of $1\,\mathrm{ppmv}^2$ molecular oxygen and water, and $1\,\mathrm{ppbv}^2$ ozone and hydrogen peroxide, $25\,\mathrm{K}^2$ for the temperature field, $25\,\mathrm{\mu T}^2$ for the magnetic field,

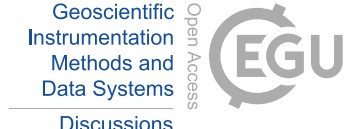

and $2500\,\mathrm{m^2\,s^{-2}}$ for the wind field. This method of defining the error covariances can be found described more by Eriksson et al. (2005). The response of the retrieved parameter to the system is computed by changing $\mathbf{e}_y$ to $\mathbf{Ju}$ in the Equation 1 above, where $\mathbf{u}$ is a vector of ones. For simplicity, each parameter is treated independently.

## 3 Results and Discussion

This section describe the forward simulations and the error retrievals. The first subsection shows the simulated signal in the frequency range we are working from. The next subsection presents the retrieval error estimates for limb observations with satellite altitude at approximately 400 km. These retrieval errors are from a one second long integration time per tangent altitude. The last subsection presents nadir retrieval errors for a satellite at approximately 3 Mars radii. Since these observations are in nadir staring mode, one full hour of integration time pointing towards the same area is assumed to reduce estimated noise.

### 3.1    Modeled Sensor Signal

The simulated signal between 486.65 to 487.65 GHz and between 475.65 to 474.65 GHz is shown in Figure 4. The signal shows two hydrogen peroxide absorption lines at 475.20 GHz and 487.20 GHz, a single molecular oxygen absorption line at 487.25 GHz, a single ozone absorption line at 487.35 GHz, and a single water absorption line near the edge at 474.69 GHz. The limb-view signal from molecular oxygen is saturated at the lowest altitude (peaks of 95 K in the double sideband view),

but its signal is much reduced at higher altitudes. Water is saturated to higher altitudes because of the constant volume mixing ratio simplification. The ozone signal is weak, at most a few Kelvin in strength at low altitudes. The hydrogen peroxide signal is also weak, but much stronger than the ozone signal, at ten Kelvin at low altitudes.

### 3.2    Limb-View Error Estimations

The estimated errors on the temperature field, wind field, and magnetic fields are shown in Figure 5. The simulations indicate

we are sensitive to the temperature to up to 80 km altitude with errors of less than 2 K, that between 30 and 70 km we can retrieve wind at less than 15 m/s, and that between 30 and 50 km we can retrieve the magnetic field at about $1\,\mu\mathrm{T}$. The temperature field results are strongly dependent on the water content. Reduce the trace gases by a factor 100 and temperature measurement sensitivity is reduced to the range between 10 and 40 km, with estimated errors about twice as large. Wind errors are worsened by a factor $20\,\%$ in altitude range and retrieval error estimation due to the reduction in trace gases. Magnetic field

retrievals, being from the polarized molecular oxygen signal, is not affected at all by changing the trace gases. In nadir-view geometry, the simulated signals are weak for all four molecules. Nadir signals must therefore be integrated for much longer.

     That temperature retrievals so strongly depend on water content will be a limiting factor in this mission. We will need other means to constrain the temperature field at higher altitudes or expect even larger uncertainties in retrieved molecular volume mixing ratios at those altitudes. Magnetic field retrievals of $1\,\mu\mathrm{T}$ could seem inadequate to improve on present knowledge.

However, it is important to note that the crustal magnetic field is going to remain the same measurement after measurement. Thus, the single profile magnetic field retrieval is not the most sensible way to judge such errors in terms of what the end



**Figure 4.** Simulated signal of our sensor. The signal unit is in Planck-equivalent brightness temperature as expected with the sensor calibrated to have half of its measured signal from each sideband. Upper row shows limb-view geometry. Legend contains tangent altitudes. Lower row shows nadir-view geometry.

product can contribute. Instead, as measurements accumulate the statistical noise of the measurements will go down. The usefulness of this depends on the final orbit inclination and if we are able to measure inside the stronger fields near 50° south; the inclination is not considered in this work but this requires an inclination greater than about 35° at 400 km satellite altitude.

The estimated errors on the volume mixing ratio of molecular oxygen, water, ozone, and hydrogen peroxide are shown
5  in Figure 6. Molecular oxygen volume mixing ratio can be be measured at about 100 ppmv below 50 km altitude — at the lowest levels, errors are estimated to be about 20 ppmv. Having less trace gases makes no difference in the molecular oxygen

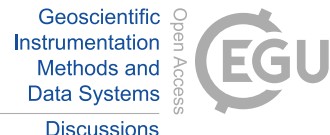



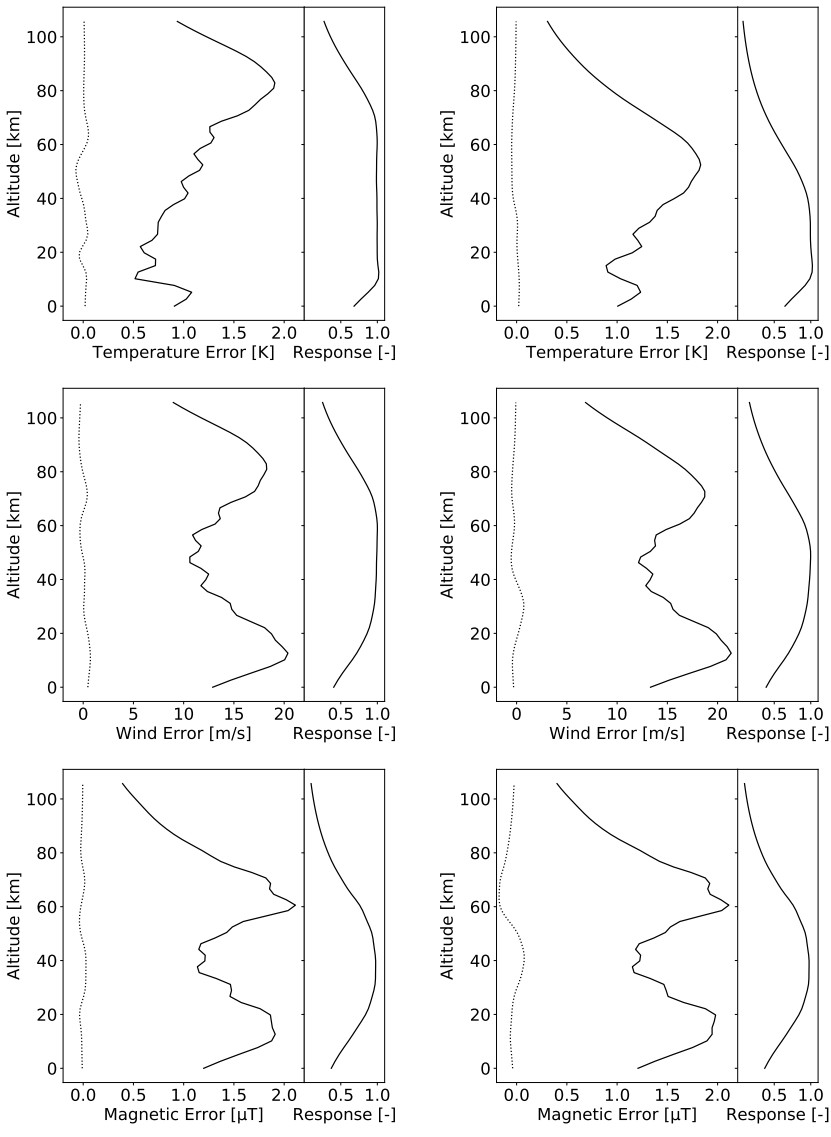

**Figure 5.** Temperature, wind, and magnetic field errors expected from a single profile of limb-view measurements. The left column of three panels are for the signal shown in Figure 4, whereas the right column of three panels are for reduced water, ozone, and hydrogen peroxide. The measurements are assumed separated by two seconds and allowed one second for signal integration time. Left column of each panel shows the estimated error. Dotted lines show the mean of 1000 realizations of Equation 1, whereas the solid lines are the standard deviation of these 1000 realizations, and therefore our estimation for the retrieval error. Right column of each panel shows the measurement response.

measurements. Water can be retrieved at better than 2 ppmv below 50 km. The lower limit of detection for water gas is at about 20 ppbv, as seen in the reduced trace gas plots. Ozone can only be retrieved to an accuracy of about 2 ppbv at altitudes near the surface. Hydrogen peroxide is better than ozone, with a retrieval limit below 1 ppbv at altitudes near the surface.





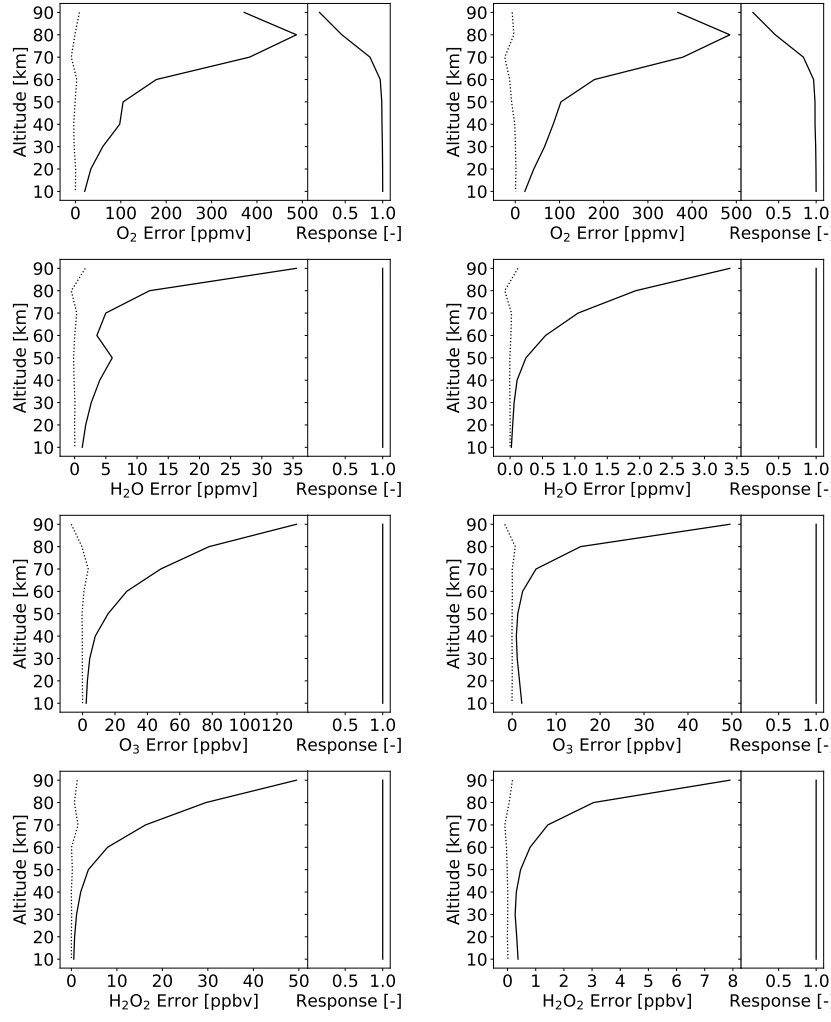

**Figure 6.** Volume mixing expected errors from a single profile in lib-view geometry. The lines and columns of the individual panels are the same as for Figure 5. The left column of four panels is for the volume mixing ratios described in the methodological section. The right column of four panels have the same molecular oxygen and carbon dioxide content but the trace gases are reduced by a factor 100, and shows the lower detection limit.

One of the key features of Mars that our sensor could potentially help with is to measure some limited dust storm-induced chemistry. Dust storms can have large electric fields, which means that there could be a large increase in hydrogen peroxide (as suggested in the works by, e.g., Delory et al., 2006; Atreya et al., 2006). Since sub-millimeter waves are barely affected by atmospheric dust, such an increase in hydrogen peroxide could become our best tracer for determining if a measurement is

5    from inside a dust storm or not. The estimated errors for molecular oxygen at low altitudes is sufficiently low to track changes on levels better than the differences measured by various missions before us (Carleton and Traub, 1972; Hartogh et al., 2010;



Mahaffy et al., 2013; Sandel et al., 2015). This means the instrument should be able to map molecular oxygen to potential daily, seasonal, and weather changes

### 3.3 Nadir-View Error Estimations

The estimated errors on the temperature field, wind field, and magnetic fields are shown in Figure 7. Temperature can be
retrieved to about a Kelvin below 40 km. In the case with low amount of trace gases, the error and altitude range remains about the same for the temperature field as with more trace gases, but the degrees of freedom are reduced. Wind retrievals of 5 m/s is possible, but the idea that Mars retains a 5 m/s vertical wind for an hour, as is required for nadir measurements to observe winds, is unlikely. So the vertical wind field is practically not retrievable. Finally, the magnetic field can be retrieved to about $0.5\,\mu T$ at 50 km altitude. This number is lower than for limb-view geometry, but it would require pointing our sensor at the
same region of Mars for the entire duration to be useful. Even if it is possible to obtain enough data from the same region after many passes to yield useful results, the spatial resolution from 3 Mars radii is relatively poor.

The volume mixing ratio errors for nadir view for normal amount of trace gases is about 40 ppmv column average for molecular oxygen, about 0.8 ppmv for water, about 5 ppbv for ozone, and about 1 ppbv for hydrogen peroxide. In the case with reduced trace gases by a factor 100, the other molecules have about the same expected errors but water is detectable down to
30 ppbv. This indicates that we could do some atmospheric chemistry and meteorology with a pure nadir-viewing sensor, but that it is much better to use as much of our time as possible in limb-viewing geometry.

### 4   Conclusions

We are planning a mission to Mars using a micro-satellite to carry a sub-millimeter sensor to orbit and have performed a feasibility study to determine what we can measure and to what success. The sensor will measure atmospheric molecular
oxygen, water, ozone, and hydrogen peroxide to retrieve their volume mixing ratios and change over time. The sensor will also retrieve the temperature field, the wind field, and the magnetic field at various levels of success. The expected measurement errors for molecular oxygen is below 100 ppmv in limb view below 50 km, with 20 ppmv for near surface measurements. For water in limb-view, the retrieval errors are below 1 ppmv with a detection limit of a few tens of ppbv. For ozone the limits are at 2 ppbv, and for hydrogen peroxide the retrieval limits are in the range of 1 ppbv. In nadir-viewing geometry, the expected
errors in the column are worse but not by much since we can keep integrating the signal from the same area for a long time, though the vertical resolution clearly suffers. We think these measurements will be helpful to better understand the Martian atmospheric molecular oxygen cycle.





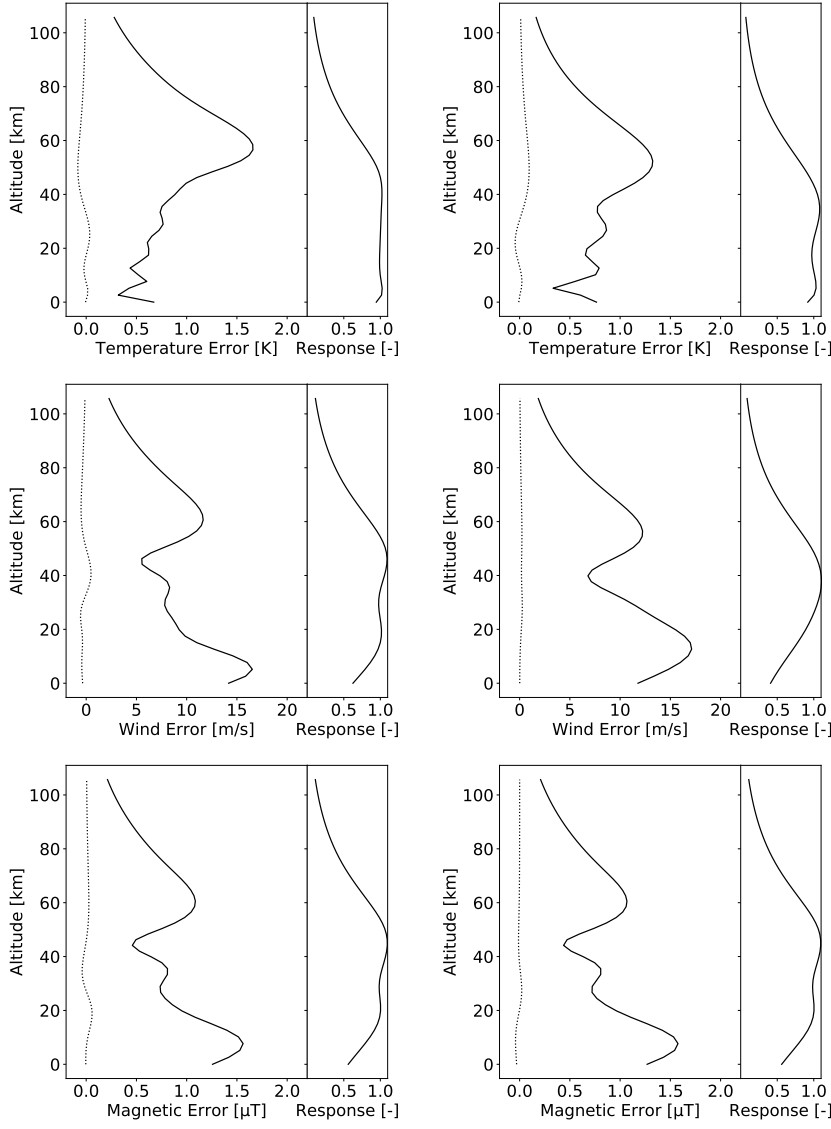

**Figure 7.** Same as Figure 5 except for nadir geometry and one hour of integration time.

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
