# Peer review of "Mars sub-millimeter sensor on micro-satellite: sensor feasibility study"

_Geoscientific Instrumentation, Methods and Data Systems, 2017_

## Referee Comment (RC1) · Anonymous Referee #1 · 5 Mar 2018

This manuscript describes an effort to evaluate the potential scientific contributions of a planned submillimeter sounder making limb and nadir observations of Mars' atmosphere from a Martian orbit. While the instrument may indeed make useful and needed contributions to the state of knowledge, I'm afraid this paper in its current form does not convey those potential contributions at all well, and it needs further work before it is in a useful state to truly convey the information needed.

The authors have overlooked a fundamental issue inherent in the formulation of atmospheric remote sounding instruments, in the algorithms used to retrieve vertical profiles of atmospheric properties from the raw radiance measurements, and in the scientific interpretation of those profiles. The issue is that it there is an underlying tradeoff between precision and resolution. That is to say, one can always improve the precision

of a measurement at the expense of coarser vertical resolution. The factors that affect that trade are varied and have different impacts. One factor that the paper touches on is the choice of the 10-km length scale in the a priori covariance matrix (page 5, line 20). Increasing this to, say, 20-km will reduce the vertical resolution of the result while improving the precision. For a limb-scanning instrument, implementing a slower vertical limb scan can improve both precision and vertical resolution but, because the instrument is moving, this 3 results in a worsening in effective horizontal resolution. Developing a lower-noise submillimeter wave receiver can lead to improvements in all properties.

The authors seem to be unaware of this trade, or at least do not discuss it in the paper. By its very nature, this trade means that it is meaningless to discuss the precision of the measurements (Figures 5, 6 and 7) without describing the resolution at the same time. On a related note, the authors fail to describe their instrument in enough detail to properly convey other aspects of this trade. The integration time is quoted as 1s, but nowhere is it stated how many limb tangent altitudes are observed and the vertical range they cover. For this paper to be suitable for publication, these key absences from the discussion must be addressed.

Related to this is the authors dwelling on "detection limits". What do they mean by that? Is it meant to be the precision in a single profile retrieval? If so, it is a poor choice of terminology for it, as that "limit" can be reduced by, for example, averaging together observations over multiple days to improve the precision (at the expense of poorer temporal resolution - another example of a precision/resolution trade). I would avoid the use of detection limit, as it is not an accurate description of the way such measurements work. They are characterized (in the traditional but perhaps simplistic view) by precision, resolution, and accuracy. Using the term detection limit implies there is some minimum abundance of a molecule in the atmosphere that is needed to somehow trigger a useful measurement. The reality however is that, for example, if the single-profile precision of the instrument is 0.1 ppmv for a species, it doesn't matter

whether the true atmospheric abundance is 0 ppmv or 10 ppmv, the "noise" in the result is the same.

A secondary weakness of this paper is that, while claiming to follow the formalism of Rodgers 2000 (and others who take that approach), the authors have applied aspects of that formalism in a rather bizarre way, while ignoring other important aspects of it. Specifically, the authors make no explicit reference to the Averaging Kernels that are key to the interpretation of such measurements (and provide the resolution information so sorely needed as described above). While their approach is not a debilitating weakness as such, it does speak of a lack of understanding by the authors that it would be better to remedy if the paper is to be viewed credibly by the community.

In updating their manuscript, I would suggest that the authors view the earlier paper on this topic by Urban and colleagues [Urban, J., K. Dassas, F. Forget, and P. Ricaud (2005), Retrieval of vertical constituents and temperature profiles from passive submillimeter wave limb observations of the Martian atmosphere: a feasibility study, Appl Optics, 44(12), 2438-2455, doi:10.1364/AO.44.002438] to be a good example of the approach that properly uses the Rodgers framework, and includes a recognition of precision/resolution trades.

My final general observations is that the standard of English in this paper is rather poor. I have tried to indicate some specific suggestions for improvement, in the spirit of improving the manuscript that eventually emerges from the authors. However, I suspect that, in the long run, it would greatly benefit from a careful scrubbing by a copy editor.

==== Specific points

—- Page 1

Abstract line 1: "We are planning" is vague. What state of readiness is this mission concept in? Is this just a proposal or a planned-proposal or is it a confirmed mission? If it's in the formulation stage, is there a specific funding agency and/or mission opportunity

being targeted.

Line 2/3: "The sensor will measure ++emission from++ atmospheric ... peroxide ++in order++ to retrieve... and ++the++ changes ++therein++ over time".

Line 4: "levels of success" is rather un-scientific. I suggest "with various degrees of precision and resolution"

Lines 5-9: Why not summarize the wind results also.

Line 9: "... the vertical resolution clearly suffers". I don't know how you can say that as the narrative makes no effort to quantify the vertical resolution as it stands

Line 11: "We are in the works to" - as with the abstract, this is very vague, please clarify along the lines discussed above.

Line 16: You fail to describe why the fact that the radiation is passively emitted is an advantage. Are you comparing to some active (e.g., radar, lidar?) sensor, or are you comparing to solar occultation (with its sparse coverage) or observations of sunlight scattered from the limb (limited to daytime conditions only).

Line 19: "have" -> "has"

—- Page 2

Line 3: Comma needed before "but" and an "are" needed after it.

Line 4: "oxidize" -> "oxidizes"

Line 5/6: This sentence is very poorly worded. I suggest something like "Sandel et al. (2015) showed that a non-constant oxygen mixing ratio profile is needed to explain solar occultation observations above 90km", assuming that's actually what their paper stated.

Line 7: "We will be able to see..." should be "The instrument we describe here will be able to see..."

Line 11/12: This is where one should talk about precision vs. resolution trades.

Section 2.2: As discussed above, you need to give more information on the instrument scan. Over what vertical range is the tangent point scanned? What is the vertical spacing of the limb views and how long does the vertical scan take. How far does the tangent point move horizontally during that scan (a measure of the effective horizontal resolution).

—- Page 4

Line 15: See the earlier discussion on the term "detection limit". Also, I completely fail to see why the simulation with 100x less $O_2$ is desired or even valid. Why focus on such an unrealistic case? I guess if the system is linear enough then the true amount doesn't matter, but then, still, why pick an unrealistic value.

—- Page 5

Section 2.4

As discussed above, the authors have taken rather a bizarre route to performing their analysis. While equation 1 is valid in itself, their discussion of it, and their "changing $e\_y$ to $Ju$" (which is related to the computation of the averaging kernel, though they do not describe it that way) is out of family with the traditional approach. The authors make no effort to explain what is meant by "the response of the retrieved parameter to the system". I take it to be the factor plotted in the right hand part of each panel in figures 5-7, and assume it's the area under each averaging kernel row, but I'm not sure. As stated above, this discussion needs to be updated to make proper reference to vertical resolution (and/or degrees of freedom for signal), ideally in the context of their averaging kernels.

Also, the authors seem to be computing the precision on Level 2 products through some kind of Monte Carlo approach (at least that's what the Figure 5 caption states, although incomplete information is given, what distribution is assumed for $e\_y$, for ex-
ample). I have no idea why the authors took that approach rather than simply computing the covariance of the precision directly using the standard approach (see equation 3.19 of the Rodgers book, page 50). This is readily derived from their equation 1.

Further, the dotted lines in Figures 5-7 are completely meaningless, as they simply represent one realization of the distribution whose standard deviation is (ignoring vertical resolution issues) given by s_x/sqrt(1000). If they'd chosen 1,000,000 realizations instead the dotted line would be closer still to zero, but what does that tell us.

Line 17: Clarify "assuming independence". I think you mean independence between the various families of terms here (wind, temperature, vmr etc.), not within the terms themselves (as you do have a vertical covariance within each species, as described on line ~20).

Line 18: Insert "radiance" between "diagonal" and "error"

—- Page 6

Line 2-3: Again, restructure this to talk about the averaging kernels instead.

Line 5: Suggest you change "error retrievals" to "precision estimation" or similar.

Line 25: "is" -> "are"

Line 26: Please quantify "much longer"

Line 28: "or expect" -> "to avoid"

—- Page 8:

Figure 5 (and 6 and 7): As discussed, replace (or augment) the right hand part of each figure with averaging kernels and/or their full width at half maximum. Dispense with the dotted lines, which convey no useful information.

Line 1 (1st line below caption): Please define what is meant by "The lower limit of detection for water [vapor not gas]"?

Line 2: Where am I supposed to get the 20 ppbv number from in these figures? Is this what you're trying to do with the dotted line? I fail to follow. Even if the dotted line meant anything useful (which it doesn't) are you seriously asking the reader to see it having a 20 ppbv value anywhere, when the ticks on the x-axis are 5 ppmv! Again, is it really a lower limit? A monthly zonal mean would be able to pick out the abundance with far better precision.

Line 3: Define "better" is this in absolute vmr or some kind of fractional sense?

—- Page 10

Line 6: This is the first time you've mention degrees of freedom, please introduce it properly.

Line 7: The discussion of a 5m/s vertical wind again implies to me that the authors do not understand that precision is not the same as "detection limit". That said, I agree with their assessment that vertical wind (which I presume is much slower than horizontal) will not be measureable from nadir.

---

## Referee Comment (RC2) · Anonymous Referee #2 · 15 May 2018

The manuscript is about using sub-millimeter radiometry to measure Martian atmosphere. In my opinion this idea is a solid one and combining the sub-millimeter instrument with a micro satellite platform follows the current trend.

However, I find some problems with this feasibility study. I do see that the main emphasis of the manuscript is on the radiometer instrument and the measurements made with it. I would still like to see the space technology details discussed more.

For example, the authors state that putting the satellite into Martian orbit has never been attempted before using atmospheric drag. The authors just assume that this can be done and assume a couple of potential orbits. But doesn't this mean that using micro satellite platform makes the mission harder to accomplish? Why not just put the instrument to a bigger platform and put it into orbit by more conventional means?

[Figure]

I also think that the description of the radiometer instrument is lacking. I did read the Kasai et al. 2012 paper and they describe the FIRE radiometer in detail. In what way the FIRE-mini differs from the FIRE instrument? Is it only size as the authors mention or the usage of two channels with different circular polarization? How about weight and power budget? Kasai et al. 2012 gives two options of FIRE radiometer; Limited science: 5 kg, 10 W & Full science: 16 kg, 40 W. If the micro satellite platform has max. weight of 100 kg then fitting in even the full science option of FIRE might do the job. So what is the advantage of FIRE-mini over FIRE?

Solar power is available less in the Martian orbit than in Earth orbit. Is the 40W power feasible using solar panels? In my opinion the small satellite size could be a limitation in this case. How about the attitude control? A probable choice would be magnetic control which also uses electric power. So what is the actual advantage of using micro satellite platform? Is it the reduced costs?

I think the Reviewer #1 has good comments about the trade offs in accuracy, resolution and precision. The Reviewer #1 also points out that making the radiometer more sensitive reduces the errors in parameter retrieval. This could be difficult since the receiver is heterodyne receiver (because of sub-millimeter wavelengths) or at least depends much on the calibration of the instrument. I suppose the calibration of the FIRE-mini will be done using 2.7K background microwave radiation as with FIRE? Even though the orbit selection might no have a big effect on those the attitude control probably will have. Magnetic attitude control, however, is not that precise.

The authors describe the forward model in chapter 2.3 and then I suppose chapter 2.4 describes how the inversion of forward model is done. I think the description is more on the conceptual level and it is not that easy to see what is actually done in the paper.

---

## Author Comment (AC1) · 23 Oct 2018

**1   General changes**

The changes to the paper between this and previous version are substantial. The major changes are as listed

- The error analysis has been redone completely based on overarching comments from both reviewers. This has been done mainly to be able to discuss what both reviewers mentioned as important, namely the trade off between precision and resolution in an added subsection at the end of section 3. This discussion is not exhaustive but we think it addresses the reviewers' concerns about the the

aforementioned trade off.

- The forward simulations have been changed by using available GCM model atmospheres and updating the Zeeman effect spectroscopy. The former was done to better represent cases with a changed correlation distance (or scale length) in the a priori covariance matrix required to discuss mentioned trade off. The latter is a more recent development where the main author found a small (a few percent) error in the Zeeman splitting for the used molecular oxygen line.

Other smaller changes have also been incorporated in the current version.

Below in red are the reviewers' comments one-by-one as copy-pasted from the original response files provided by Copernicus. The one alteration that the main author has done to these comments is to add appropriate LATEX-coding for references to variables in equations.

Before going over the detailed response, we wish to thank the reviewers because their suggestions have improved this work substantially.

Attached is the updated paper.

**2 Response to reviewer 1**

**2.1 General comments**

*This manuscript describes an effort to evaluate the potential scientific contributions of a planned submillimeter sounder making limb and nadir observations of Mars' atmosphere from a Martian orbit. While the instrument may indeed make useful and needed contributions to the state of knowledge, I'm afraid this paper in its current form does*

*not convey those potential contributions at all well, and it needs further work before it is in a useful state to truly convey the information needed.*

We have updated the manuscript as a response to this comment as detailed below.

*The authors have overlooked a fundamental issue inherrent in the formulation of atmospheric remote sounding instruments, in the algorithms used to retrieve vertical profiles of atmospheric properties from the raw radiance measurements, and in the scientific interpretation of those profiles. The issue is that it there is an underlying tradeoff between precision and resolution. That is to say, one can always improve the precision of a measurement at the expense of coarser vertical resolution. The factors that affect that trade are varied and have different impacts. One factor that the paper touches on is the choice of the 10-km length scale in the a priori covariance matrix (page 5, line 20). Increasing this to, say, 20-km will reduce the vertical resolution of the result while improving the precision. For a limb-scanning instrument, implementing a slower vertical limb scan can improve both precision and vertical resolution but, because the instrument is moving, this 3 results in a worsening in effective horizontal resolution. Developing a lower-noise submillimeter wave receiver can lead to improvements in all properties.*

We agree that there is a trade off between resolution and precision that was ignored in the last iteration. The current iteration should sufficiently explain this by updating both method and results to give room for such a discussion. We also agree that it important to keep developing newer and better receiver technology, and if these were available within our mass/power/cost/political budgets, we would be using them. It is beyond this work to develop new receivers.

We have updated the paper to detail what a change in length scale / correlation distance does to the retrieval errors at the end of the results section. We have also updated the paper to use somewhat less arbitrarily chosen variances. Both of these combined means the results are changed somewhat. While updating the discussion

about correlation distances, we also changed the atmospheric scenario to be more inline with GCMs. This means our magnetic field errors are greatly increased in this version.

We still keep the same integration times as well as system noise temperature in the updated calculations. We add at the end of the error analysis subsection, a discussion parroting the reviewer's point that there is a trade-off between spatial/temporal resolution and precision.

*The authors seem to be unaware of this trade, or at least do not discuss it in the paper. By its very nature, this trade means that it is meaningless to discuss the precision of the measurements (Figures 5, 6 and 7) without describing the resolution at the same time. On a related note, the authors fail to describe their instrument in enough detail to properly convey other aspects of this trade. The integration time is quoted as 1s, but nowhere is it stated how many limb tangent altitudes are observed and the vertical range they cover. For this paper to be suitable for publication, these key absences from the discussion must be addressed.*

It was an oversight to not clearly state that we are using the same scanning altitudes as in the figure showing the signal at said altitudes. Relevant parts of the text has been updated to reflect this.

The trade off between resolution and precision is now discussed as part of the correlation distance, and mentioned elsewhere in the text where deemed appropriate.

*Related to this is the authors dwelling on "detection limits". What do they mean by that? Is it meant to be the precision in a single profile retrieval? If so, it is a poor choice of terminology for it, as that "limit" can be reduced by, for example, averaging together observations over multiple days to improve the precision (at the expense of poorer temporal resolution - another example of a precision/resolution trade). I would avoid the use of detection limit, as it is not an accurate description of the way such measurements work. They are characterized (in the traditional but perhaps simplistic*

*view) by precision, resolution, and accuracy. Using the term detection limit implies
there is some minimum abundance of a molecule in the atmosphere that is needed to
somehow trigger a useful measurement. The reality however is that, for example, if
the single-profile precision of the instrument is 0.1 ppmv for a species, it doesn't matter
whether the true atmospheric abundance is 0 ppmv or 10 ppmv, the "noise" in the result
is the same.*

We have removed discussions of detection limits. While we still think single profile
detection are important (since this is interesting for short timescales), as the reviewer
notes, this kind of limit is anyways implied by the retrieval errors so introducing unclear
terms is not desired.

We still keep the reduced atmospheric gases case in the paper because the VMR
"noise" is nonlinear as the opacity gets too high for the variable species.

*A secondary weakness of this paper is that, while claiming to follow the formalism of
Rodgers 2000 (and others who take that approach), the authors have applied aspects
of that formalism in a rather bizarre way, while ignoring other important aspects of it.
Specifically, the authors make no explicit reference to the Averaging Kernels that are
key to the interpretation of such measurements (and provide the resolution information
so sorely needed as described above). While their approach is not a debilitating weak-
ness as such, it does speak of a lack of understanding by the authors that it would be
better to remedy if the paper is to be viewed credibly by the community.*

We have made the implicit references to the averaging kernel more explicit.

*In updating their manuscript, I would suggest that the authors view the earlier paper
on this topic by Urban and colleagues [Urban, J., K. Dassas, F. Forget, and P. Ricaud
(2005), Retrieval of vertical constituents and temperature profiles from passive sub-
millimeter wave limb observations of the Martian atmosphere: a feasibility study, Appl
Optics, 44(12), 2438-2455, doi:10.1364/AO.44.002438] to be a good example of the
approach that properly uses the Rodgers framework, and includes a recognition of*

*precision/resolution trades.*

It was an oversight to not cite Urban et al.'s work in previous iteration. This has been corrected.

*My final general observations is that the standard of English in this paper is rather poor. I have tried to indicate some specific suggestions for improvement, in the spirit of improving the manuscript that eventually emerges from the authors. However, I suspect that, in the long run, it would greatly benefit from a careful scrubbing by a copy editor.*

2.2   Specific points

*Abstract line 1: "We are planning" is vague. What state of readiness is this mission concept in? Is this just a proposal or a planned-proposal or is it a confirmed mission? If it's in the formulation stage, is there a specific funding agency and/or mission opportunity being targeted.*

The state of readiness now is that components are being built. The building of key parts of the platform and of the instrument is funded. It is not a confirmed mission, however, as we are still trying to find a partner to piggy-back to space with. It is likely, also, that the first iteration of this mission will attempt landing with the same instrument as we describe. The text has been updated to say the instrument is being built.

*Line 2/3: "The sensor will measure ++emission from++ atmospheric ... peroxide ++in order++ to retrieve... and ++the++ changes ++therein++ over time".*

Fixed. Thanks!

*Line 4: "levels of success" is rather un-scientific. I suggest "with various degrees of precision and resolution"*

Adopted. Thanks!

*Lines 5-9: Why not summarize the wind results also.*

Done. Thanks!

*Line 9: "... the vertical resolution clearly suffers". I don't know how you can say that as the narrative makes no effort to quantify the vertical resolution as it stands*

Removed. Thanks!

*Line 11: "We are in the works to" - as with the abstract, this is very vague, please clarify along the lines discussed above.*

Done. Thanks!

*Line 16: You fail to describe why the fact that the radiation is passively emitted is an advantage. Are you comparing to some active (e.g., radar, lidar?) sensor, or are you comparing to solar occultation (with its sparse coverage) or observations of sunlight scattered from the limb (limited to daytime conditions only).*

We added that the measurements are "independent of the local time", which, e.g., solar occultation measurements are not, as the reviewer points out.

*Line 19: "have" → "has"*

Fixed. Thanks!

*—- Page 2*

*Line 3: Comma needed before "but" and an "are" needed after it.*

Fixed. Thanks!

*Line 4: "oxidize" → "oxidizes"*

Fixed. Thanks!

*Line 5/6: This sentence is very poorly worded. I suggest something like "Sandel et al. (2015) showed that a non-constant oxygen mixing ratio profile is needed to explain*

*solar occultation observations above 90km", assuming that's actually what their paper stated.*

Updated. Their work does not show variation. It shows the VMR is higher at higher altitudes than at lower altitudes. Our intention was to point out that at some altitude range between ground and 90 km, the ratio are increasing, while measurements closer to ground are more similar. The GCM model we get VMR from does not have this increase, and does not reproduce ground measurements either. Thanks!

*Line 7: "We will be able to see..." should be "The instrument we describe here will be able to see..."*

Updated. Thanks!

*Line 11/12: This is where one should talk about precision vs. resolution trades.*

We have reduced the complexity of the retrieval simulation further by removing any correlation distances from the base simulations. In a short note on precision versus resolution at the end we bring this back into the fold, but we do not think it warrants mentioning in the introduction so have left the text unchanged.

*Section 2.2: As discussed above, you need to give more information on the instrument scan. Over what vertical range is the tangent point scanned? What is the vertical spacing of the limb views and how long does the vertical scan take. How far does the tangent point move horizontally during that scan (a measure of the effective horizontal resolution).*

We have added this to the results section. Thanks!

*—- Page 4*

*Line 15: See the earlier discussion on the term "detection limit". Also, I completely fail to see why the simulation with 100x less O2 is desired or even valid. Why focus on such an unrealistic case? I guess if the system is linear enough then the true amount*

[Figure]

*doesn't matter, but then, still, why pick an unrealistic value.*

Term removed. To clarify, we never used 100X times less $O_2$, this was an unfortunate formulation. The retrieval system is not linear for the other species but our statements were to broad regarding how we set up the simulations (it should have read "trace gases bar $O_2$" before, and the updated text takes this into account). Thanks!

*—- Page 5*

*Section 2.4*

*As discussed above, the authors have taken rather a bizarre route to performing their analysis. While equation 1 is valid in itself, their discussion of it, and their "changing $e_y$ to $Ju$" (which is related to the computation of the averaging kernel, though they do not describe it that way) is out of family with the traditional approach. The authors make no effort to explain what is meant by "the response of the retrieved parameter to the system". I take it to be the factor plotted in the right hand part of each panel in figures 5-7, and assume it's the area under each averaging kernel row, but I'm not sure. As stated above, this discussion needs to be updated to make proper reference to vertical resolution (and/or degrees of freedom for signal), ideally in the context of their averaging kernels.*

We now define the averaging kernel. We also add degrees of freedom to the plots. We further define the measurement response and what we consider a 'good' response. Thanks!

*Also, the authors seem to be computing the precision on Level 2 products through some kind of Monte Carlo approach (at least that's what the Figure 5 caption states, although incomplete information is given, what distribution is assumed for $e_y$, for example). I have no idea why the authors took that approach rather than simply computing the covariance of the precision directly using the standard approach (see equation 3.19 of the Rodgers book, page 50). This is readily derived from their equation 1. Further,*

*the dotted lines in Figures 5-7 are completely meaningless, as they simply represent one realization of the distribution whose standard deviation is (ignoring vertical resolution issues) given by $s_x/\sqrt{1000}$. If they'd chosen 1,000,000 realizations instead the dotted line would be closer still to zero, but what does that tell us.*

Yes, we realize the equation multiple times. The dotted lines have been removed. Thanks!

*Line 17: Clarify "assuming independence". I think you mean independence between the various families of terms here (wind, temperature, vmr etc.), not within the terms themselves (as you do have a vertical covariance within each species, as described on line âĹij20).*

Yes. Thanks!

*Line 18: Insert "radiance" between "diagonal" and "error"*

Done. Thanks!

*—- Page 6*

*Line 2-3: Again, restructure this to talk about the averaging kernels instead.*

The subsection is changed greatly at this point. Hopefully the reviewer agrees it is improved. Due to this, language corrections below are left without a response from us, while we apply the terminological suggestions.

*Line 5: Suggest you change "error retrievals" to "precision estimation" or similar.*

We change to use therms like "error estimations" or "expected errors" instead of "error retrievals".

*Line 25: "is" → "are"*

*Line 26: Please quantify "much longer"*

We only discuss changing integration time for $O_2$ now, and we quantify the time necessary for achieving the discussed 'targets'.

*Line 28: "or expect" → "to avoid"*

*—- Page 8:*

*Figure 5 (and 6 and 7): As discussed, replace (or augment) the right hand part of each figure with averaging kernels and/or their full width at half maximum. Dispense with the dotted lines, which convey no useful information.*

Done. Thanks!

*Line 1 (1st line below caption): Please define what is meant by "The lower limit of detection for water [vapor not gas]"?*

Water should read gaseous water throughout the text now.

*Line 2: Where am I supposed to get the 20 ppbv number from in these figures? Is this what you're trying to do with the dotted line? I fail to follow. Even if the dotted line meant anything useful (which it doesn't) are you seriously asking the reader to see it having a 20 ppbv value anywhere, when the ticks on the x-axis are 5 ppmv! Again, is it really a lower limit? A monthly zonal mean would be able to pick out the abundance with far better precision.*

We agree this was bad. The numbers we present in the current version are hopefully clearer to see now.

*Line 3: Define "better" is this in absolute vmr or some kind of fractional sense?*

*—- Page 10*

*Line 6: This is the first time you've mention degrees of freedom, please introduce it properly.*

This is added to the error analysis subsection.

*Line 7: The discussion of a 5m/s vertical wind again implies to me that the authors*

*do not understand that precision is not the same as "detection limit". That said, I agree with their assessment that vertical wind (which I presume is much slower than horizontal) will not be measureable from nadir.*

Please also note the supplement to this comment:
https://www.geosci-instrum-method-data-syst-discuss.net/gi-2017-50/gi-2017-50-AC1-supplement.pdf

**Supplement:**

**Mars sub-millimeter sensor on micro-satellite: sensor feasibility study**

Richard Larsson[1,6], Yasuko Kasai[1], Takeshi Kuroda[1], Shigeru Sato[1], Takayoshi Yamada[1],
Hiroyuki Maezawa[2], Yutaka Hasegawa[3], Toshiyuki Nishibori[4], Shinichi Nakasuka[5], and Paul Hartogh[6]

[1]National Institute of Information and Communications Technology, Tokyo, Japan
[2]Osaka Prefecture University, Osaka, Japan
[3]Institute of Space and Astronautical Science, Japanese Aerospace Exploration Agency, Tokyo, Japan
[4]Research and Development Directorate, Japanese Aerospace Exploration Agency, Tokyo, Japan
[5]Tokyo University, Tokyo, Japan
[6]Max Planck Institute of Solar System Research, Göttingen, Germany

*Correspondence to:* Richard Larsson (larsson@mps.mpg.de)

**Abstract.** We present a feasibility study for a sub-millimeter instrument on a small Mars platform now under construction. The sensor will measure the emission from atmospheric molecular oxygen, water, ozone, and hydrogen peroxide in order to retrieve their volume mixing ratios and the changes therein over time. In addition to these, the instrument will be able to limit the crustal magnetic field, and retrieve temperature and wind speed with various degrees of precision and resolution. The expected measurement errors before spatial and temporal averaging are 15 ppmv to 25 ppmv for the molecular oxygen mixing ratio, 0.2 ppmv for the gaseous water mixing ratio, 2 ppbv for the hydrogen peroxide mixing ratio, 2 ppbv for the ozone mixing ratio, $1.5\,\mathrm{\mu T}$ to $2.5\,\mathrm{\mu T}$ for the magnetic field strength, $1.5\,\mathrm{K}$ to $2.5\,\mathrm{K}$ for the temperature profile, and $20\,\mathrm{m\,s^{-1}}$ to $25\,\mathrm{m\,s^{-1}}$ for the horizontal wind speed.

**1 Introduction**

We are building a sub-millimeter sensor with aim towards Mars using a micro-satellite platform. The goal is to launch in 2024, and the main scientific target is to measure changes in Martian molecular oxygen over time. The sensor is based on a previous proposal of a more advanced version of the instrument by Kasai et al. (2012), and our working name for the sensor and platform is the Terahertz Experiment (TEREX). As both Urban et al. (2005) and Kasai et al. (2012) has previously suggested, the advantages of sub-millimeter technology for Martian remote sensing is that the radiation at sub-millimeter frequencies are mostly unaffected by atmospheric dust content, and that the radiation observed is passively emitted by the target atmospheric gases. The sensor will be able to measure molecular oxygen, gaseous water, hydrogen peroxide, ozone, the temperature field, the wind field, and the strongest crustal magnetic fields. These observations are possible from radiation emitted from within dust storms, and are independent of the local time.

Martian molecular oxygen has been measured several times by missions like Viking, Herschel, Curiosity, and MAVEN. Carleton and Traub (1972) measured a global molecular oxygen profile of 1300 ppmv, Hartogh et al. (2010) measured a constant 1400 ppmv profile in whole disk measurements but remarked that there could be a higher concentration near the surface,

Mahaffy et al. (2013) measured a constant profile of 1450 ppmv in ground-based measurements, and Sandel et al. (2015) measured up to 4000 ppmv at altitudes of 90 to 120 km in limb measurements at nanometer wavelengths. The discrepancies are small between most of these, but are important and should be studied in more details. Molecular oxygen acts as the chemical background that oxidizes gaseous water and various hydrogen radicals. Low altitude variations have not been measured in detail over time. The Sandel et al. (2015) results shows that the profile cannot be constant from the ground up to 90 km, so the volume mixing ratio must increase in some altitude range. Our sensor will not be able to confirm these measurements because the altitude range of Sandel et al. (2015) is above our reach. The instrument we describe here will be able to see if the increase observed at 90 km are reflected near the surface, and limit the altitude range at which the relative oxygen concentration starts increasing. Another of the key features of Mars that our sensor can help with is to measure some limited dust storm-induced chemistry. Dust storms can have large electric fields, which means that there will be a large increase in hydrogen peroxide when these storms are active (as suggested in the works by, e.g., Delory et al., 2006; Atreya et al., 2006). Since sub-millimeter waves are barely affected by atmospheric dust, the instrument will be able to sense hydrogen peroxide production inside the dust cloud.

This paper is dedicated to show the feasibility study we performed to test the sensor design of TEREX. There will be other papers about the mission to describe the details of the orbit insertion and retention, and to give a broader scientific overview of the mission as a whole. This paper shows the forward simulations of the expected observations, and the error estimations from a simple retrieval setup. The next section describes the method of how we set up our simulations — it also discusses some of the limitations that we have encountered. After the description of our method, we show our results, discuss their consequences, and give our concluding remarks.

**2  Method**

This section goes through the basic assumptions we made to perform the simulations. This includes orbit considerations, sensor design, spectroscopic modeling, and retrieval procedure.

**2.1  Orbit**

The sub-millimeter sensor will be carried to Mars on a satellite weighing less than 100 kg. Such a small satellite needs special means to enter orbit. The selected method to achieve orbit is to use the atmospheric drag to perform aerocapture. Aerocapture has never been attempted successfully before — as far as we are aware — so final orbital parameters are to some extent uncertain. A future work will discuss the orbit insertion and retention in details. For this work, we have opted to work simply with two sets of observations taken from an orbit with the Kepler elements semi-major axis of $6150\,\mathrm{km}$ and eccentricity of 0.5. This gives a periareion altitude of $400\,\mathrm{km}$ and an orbit time of 5 hours and 20 minutes. We will show simulated observations as well as estimated errors for retrieved atmospheric quantities in limb scanning mode near the periareion and in nadir staring mode near the apoareion. The final orbit could differ from the one above, but the feasibility of the sensor is not strictly dependent on the details of the orbit. As long as the periareion is not too high, it will merely affect its spatial resolution.

**2.2 Sensor**

The sensor is also limited mass and power budget by the small scale of the platform itself. We will for instance not be able to change the local oscillator to be sensitive to different ranges than those selected on the design board. A schematic sketch of the instrument can be seen in Figure 1. The spectrometer we plan to use is the chirp transform spectrometer designed for the Jupiter icy moons explorer's sub-millimeter wave instrument, with 10000 channels over a 1-GHz range (for examples on this type of spectrometer, see, e.g., Hartogh, 1997). The local oscillator will be at 481.15 GHz with a central intermediate frequency of 6 GHz. There will be no suppression of either of the sidebands, so the measured radiation will be from between 474.65-475.65 GHz and 486.65-487.65 GHz. The system noise temperature is expected to be about 2000 K in double-sideband mode. The antenna is planned to be a 30 cm large and made of carbon fiber reinforced plastic. This achieves a 10 km vertical footprint at orbit altitudes below about 400 km, with a resolution half-width of about 0.14°. A lower system noise would improve all results presented below, though the quoted number is as good as we expect the instrument will be at time of flight.

[Figure]

**Figure 1.** Sketch of sensor electronics design. The radio frequencies (RF) between 474.65-475.65 GHz and 486.65-487.65 GHz enters from the left in the figure and is split by a circular polarizer (CP) into left-handed, and right-handed, circular polarization (LHCP, RHCP) as demonstrated in the colored plots above. The signal is then mixed with the signal of a local oscillator at 481.15 GHz, to lower the frequency to a measurable range. At an intermediate frequency of 6 GHz the chirp transform spectrometer turns this analog signal into a digital signal that is fed into a computer for preparations and to be sent back to Earth.

We plan to use two identical receiver systems with the same setups but fed different states of polarization. The polarization states will be separated using a feed horn antenna. This way, if one of the receiver chains experience technical issues, we can use the other to keep measuring the atmosphere. In addition, while both chains work we can measure the magnetic field.

Molecular oxygen is affected by the Zeeman effect (Zeeman, 1897), so it is possible to sense the strongest crustal magnetic field through the state of polarization (for more details, please see Larsson et al., 2013, 2014, 2017; Larsson et al.).

Because of the dual receiver systems, if we take a single measurement every second, the measurements alone produces about 160 kbps of data before any compression is applied. This exceeds our maximum transfer rate, so we will have to perform limited data reduction on-board the spacecraft. The software for this is not ready yet. The only reduction considered in this work is an averaging by pairing immediate neighboring channels to increase the effective width of a channel to 200 kHz. This is still well below the line width of the absorption lines that we consider here, so no physics is lost.

**2.3 Forward Model**

We use the Atmospheric Radiative Transfer Simulator (ARTS; Eriksson et al., 2011; Buehler et al., 2018) for all forward simulations. ARTS is a fully three-dimensional model with full polarization capabilities that have been used in numerous studies. Please see the two cited articles, other articles citing them, and the source code — available via a copyleft license at www.radiativetransfer.org — to understand the radiative transfer method of ARTS in more details. All of the data used in this study can be found via aforementioned link.

The standard scenario atmosphere is from daytime simulations by Laboratoire de Météorologie Dynamique's global circulation model (Forget et al., 1999) for a solar angle of zero degrees. The temperature and volume mixing ratio profiles for key species are shown in Figure 2. The magnetic field is set to a constant $1\,\mu T$ throughout the transfer, pointing along the line of sight of the transfer. We expect there to be a large variations in some gases (i.e., gaseous water, ozone and hydrogen peroxide), and in the strength of the magnetic field. In order to demonstrate how the error estimation changes as the forward model scenario changes, we perform the simulations at a factor 100 times less volume mixing ratio of these gases and of the magnetic field. In this reduced case, we still keep the temperature and molecular oxygen mixing ratios the same as in the standard scenario. We do not set the wind speed in any scenario but perform retrievals of it from analytical expressions of its Jacobian.

A spectroscopic suite suitable for Mars was developed by Buehler et al. (2018), which we use for our simulations. It computes pressure broadening and shifting per atmospheric species rather than from a predefined atmospheric composition. We also use the carbon dioxide collision-induced absorption from Gruszka and Borysow (1997) as a continua shown in Figure 3. This continua is key to low altitude limb measurements, and is the main reason we are focusing on the $400\,\mathrm{GHz}$ range rather than higher frequencies. Note that the continua is effectively twice as strong as normal since we will not suppress the lower or upper sidebands. A big problem with this continua is that it is only defined down to a temperature of $200\,\mathrm{K}$. For lower temperatures we simply extrapolate to these using the Gruszka and Borysow (1997) code by ignoring the warnings. Despite these issues, we still choose to include the continua since ignoring a potential 20-120 K signal would be catastrophic for the science of the mission. Clearly, the carbon dioxide continua has to be studied more for Mars atmospheric conditions since its influence is notably strong. Even if our extrapolation is the cause of most of the absorption that we see (because of some model artifact at lower temperatures), it is necessary to confirm that this is the case, and to limit the influence of the continua.

[Figure]

**Figure 2.** Temperature and volume mixing ratio profiles used in the simulations.

[Figure]

**Figure 3.** Limb pencil view simulation of Mars considering only the carbon dioxide continua. Pencil tangent altitudes are indicated by the legend. The increase in the signal at higher altitudes and higher frequency is probably a consequence of the extrapolation of the model parameters.

**2.4 Error Analysis**

We use the optimal estimation method described by Rodgers (2000) to predict at what level of precision and resolution we will be able to measure atmospheric parameters. This is done from assuming a linear error. Errors tend to be linear near the target even if the retrieval process or underlying physics is non-linear. The computed error is from

$$\mathbf{e}_x = \mathbf{S}_a \mathbf{J}^\top \left( \mathbf{J} \mathbf{S}_a \mathbf{J}^\top + \mathbf{S}_e \right)^{-1} \mathbf{e}_y, \tag{1}$$

where $\mathbf{S}_a$ is the a priori covariance matrix, $\mathbf{J}$ is the Jacobian matrix, $\mathbf{S}_e$ is a diagonal radiance error covariance matrix, and $\mathbf{e}_y$ is a Gaussian realization of the measurement error. This realization is repeated a number of times to estimate the error on retrieved parameters. The error covariance is set to be diagonal, having the square of the standard deviation of the measurement error at each point. For most of the simulations below we assume a diagonal a priori covariance matrix. For the one figure we do not

at the end, the correlation of this matrix is assumed to decrease exponentially the further the distance is from the diagonal, as described by Eriksson et al. (2005), and the correlation between retrieved parameters (i.e., wind, temperature, magnetic field strength, $O_2$, $H_2O$, $H_2O_2$, and $O_3$ volume mixing ratios) are still assumed to be zero. The a priori variance at each altitude level is from a standard deviation of the molecular oxygen mixing ratio of 100 ppmv, of the gaseous water mixing ratio of 1 ppmv, of the hydrogen peroxide and ozone mixing ratios of 10 ppbv, of the magnetic field strength of $10\,\mu T$, of the temperature of $10\,K$, and of the wind speed of $100\,ms^{-1}$. These numbers were chosen because they are the closest order of magnitude standard deviations that produces a clear measurement response for each model parameter in the limb scanning simulations at $1\,s$ integration time. For some of these parameters, we will discuss what happens to the error estimations by increasing the standard deviation by one order of magnitude. The measurement response is defined as

$$\mathbf{r} = \mathbf{S}_a \mathbf{J}^\top \left( \mathbf{J} \mathbf{S}_a \mathbf{J}^\top + \mathbf{S}_e \right)^{-1} \mathbf{J} \mathbf{u} = \mathbf{A} \mathbf{u}, \tag{2}$$

where $\mathbf{u}$ is a vector of ones, and $\mathbf{A}$ is the averaging kernel. We define a clear measurement response as $\max(\mathbf{r}) > 0.8$. The trace of $\mathbf{A}$ gives the degrees of freedom of the retrieval system.

Note that the shape of the averaging kernels gives the altitude range where the measurements are sensitive to the atmospheric parameters, a combination of the instrument's statistical and physical vertical resolution. The degrees of freedom of the averaging kernel gives a rough estimation of how many distinct parameters can be set over the entire profile altitude range, given the statistical constraints. A reduction in the degrees of freedom by, e.g., assuming a larger statistical correlation distance or by integrating the signal over a wider vertical slice of the atmosphere, should be followed by an increase in the precision of the retrieved parameter at the cost of a loss in vertical resolution. It is also possible to increase the precision of the retrieved model parameters by increasing the integration time to reduce the measurement noise. This has the cost of reducing the spatial resolution either vertically or horizontally. We will not consider this type of reduced resolution in this work, partly because such a reduction can be done by averaging measurements a posteriori so long as the original data is available in a high time resolution. The only test of resolution versus precision we make in this work is one where we vary the a priori correlation distance. Although a very crude estimation, it serves the purpose of finding a limit to the precision.

**3 Results and Discussion**

This section gives the forward simulation results and the estimated errors on the retrieved parameters. The first subsection shows the simulated signal in the frequency range we are working in. The next subsection presents the error estimates for limb scanning observations with a satellite altitude at 400 km for the presented signal and for a signal with 100 times less gaseous water, hydrogen peroxide, and ozone, and with a 100 times weaker magnetic field. The error estimations are from a one second long integration time per tangent altitude with all tangent profile used to make up the inputs to Equation 1 for the error estimations. The next subsection also presents nadir staring error estimations. We set the integration time to one hour in nadir staring mode to reduce the signal to noise ratio.

**3.1 Modeled Sensor Signal**

The simulated signal from $486.65\,\text{GHz}$ to $487.65\,\text{GHz}$ and from $475.65\,\text{GHz}$ to $474.65\,\text{GHz}$ is shown in Figure 4. The signal shows two hydrogen peroxide absorption lines at 475.20 GHz and 487.20 GHz, a single molecular oxygen absorption line at 487.25 GHz, a single ozone absorption line at 487.35 GHz, and a single gaseous water absorption line near the edge at 474.69 GHz. The limb view signal from molecular oxygen is saturated at the lowest tangent altitude (peaks of 95 K in the double sideband view), but its signal is much reduced at higher altitudes. The gaseous water signal is also saturated at lower tangent altitudes but weakens greatly at higher tangent altitudes. The ozone signal is weak, about 5 K in strength at low altitudes but remains about as strong at higher altitudes due to its by altitude increasing volume mixing ratio. The hydrogen peroxide signal is also weak, but much stronger than the ozone signal, at 10 K at low altitudes, but its strength decreases more rapidly at higher altitudes.

**3.2 Error Estimations**

The estimated errors on the forward model parameters for limb scanning observations are shown in Figures 5 for a single second of integration time pointing at each of the tangent altitudes of Figure 4. In addition to the signal in Figure 4, we also present a case in Figure 6 where the gaseous water, the hydrogen peroxide, and the ozone volume mixing ratios, as well as the magnetic field strength, are reduced by a factor 100. Below, for simplicity, the atmosphere with less water is called 'dry' and the atmosphere with more water is called 'wet'. The estimated errors on the forward model parameters for nadir staring mode are shown in Figure 7 for one full hour of integration time.

About Figure 5, Figure 6, and Figure 7,

– Molecular oxygen: Expected error levels are of 25 ppmv with good measurement response from $10\,\text{km}$ to $30\,\text{km}$ in limb scanning mode. In nadir staring mode, the altitude range of good measurement response extends from $10\,\text{km}$ to $40\,\text{km}$, and the expected error is about 15 ppmv. Molecular oxygen is not expected to change at short timescales, so either scenario is good for the purpose of the mission to limit molecular oxygen variations. Longer integration times in limb scanning mode, or staring at lower altitudes, could be useful for limiting the vertical profile of the molecule better, especially at lower altitudes. For instance, the very near-surface error achieve good measurement response by merely increasing the integration time to $2\,\text{s}$ staring every $10\,\text{km}$ from $5\,\text{km}$ to $35\,\text{km}$.

– Gaseous water: Reducing the gaseous water content increases sensitivity at lower altitudes as the line absorption is no longer saturated. In limb scanning mode, the expected error is around 0.2 ppmv, with a range of good measurement response from $40\,\text{km}$ to $70\,\text{km}$ for a wet atmosphere and from $10\,\text{km}$ to $80\,\text{km}$ in a dry atmosphere. In nadir staring mode, the measurement response is good from $10\,\text{km}$ to $40\,\text{km}$ with similar levels of error estimated as for the limb scanning mode. To have a good measurement response near the surface for a single profile in a wet atmosphere, the a priori standard deviation of water can be increased by an order of magnitude, which yields an error estimation of about 3 ppmv and good measurement response from ground to $40\,\text{km}$.

[Figure]

**Figure 4.** Simulated signal for the sensor. The signal unit is in Planck-equivalent brightness temperature as expected with the sensor calibrated to have half of its measured signal from each sideband. The upper row shows the simulated limb scanning geometry for several tangent altitudes (used in later figures). The legend contains the tangent altitudes. The lower row shows the simulated nadir staring mode signal (also used in later figures).

– Hydrogen peroxide: In limb scanning mode, hydrogen peroxide has expected errors around 2 ppbv in both wet and dry atmospheres. Also the good measurement response altitude range is from $10\,\text{km}$ to $80\,\text{km}$ in both scenarios. The estimated error is similar in the nadir staring mode, but the altitude range is reduced from $10\,\text{km}$ to $40\,\text{km}$. An estimated error of 2 ppbv is about $20\,\%$ of the total hydrogen peroxide content in the standard wet atmosphere. Since the species is expected to increase by dust storms, this error should be good enough to characterize its production and destruction inside said storms.

[Figure]

**Figure 5.** Forward model parameter error estimations for the atmosphere of Figure 2 in limb scanning mode with $1\,\mathrm{s}$ integration time at each of the tangent altitudes of Figure 4. By columns, the left column shows the total error estimations and these errors, the central column shows the measurement response by the solid line with the dashed lines marking the 0.8 level, and the right column shows the transpose of the averaging kernel, color coding its columns and showing its trace or the degrees of freedom of the setup. The rows show the forward model parameters whose retrieval are being simulated, with the top row showing molecular oxygen, the second row showing gaseous water, the third row showing hydrogen peroxide, the fourth row showing ozone, the fifth row showing magnetic field, the sixth row showing temperature, and the last row showing wind speed.

– Ozone: The instrument is not sensitive to ozone in nadir staring mode. In limb scanning mode, the error is expected to be about 2 ppbv. For the wet atmosphere, the good measurement response range is limited to from $10\,\mathrm{km}$ to $30\,\mathrm{km}$. The dry atmosphere offers a much greater responsive measurement range from $10\,\mathrm{km}$ to $60\,\mathrm{km}$. The wet atmosphere scenario has about 50 ppbv in an altitude range around $50\,\mathrm{km}$, so an error at 2 ppbv is good enough to see variations at high altitudes.

[Figure]

**Figure 6.** Same as Figure 5 but with gaseous water, hydrogen peroxide, and ozone lowered by a factor 100 to the atmosphere of Figure 2 and a magnetic field strength of $10\,\text{nT}$. Also, molecular oxygen is left out of this figure because it is not affected by the change.

Closer to the surface the ozone mixing ratio increases to above 100 ppbv, but our instrument will only be sensitive for a single profile measurement at those altitudes if the atmosphere is dry and we keep the current constraints. It is possible to be sensitive, e.g., by increasing the standard deviation of ozone in the a priori matrix by one order of magnitude. Doing that, the estimate error increase to about 25 ppbv and near-surface altitudes have good measurement response.

5      – Magnetic field strength: The error is not affected by the wetness of the atmosphere. For the limb scanning mode scenario, the estimated error is about $2.5\,\mu\text{T}$ in the altitude range from $20\,\text{km}$ to $60\,\text{km}$. In nadir staring mode the error is expected at $1.5\,\mu\text{T}$ from $20\,\text{km}$ to $50\,\text{km}$. Note that the magnetic field is not changing over time but is from sources frozen into the crust. So many subsequent orbits can be used to limit the strength of the magnetic field more strongly. The sensor

[Figure]

**Figure 7.** Same as Figure 5 but for nadir staring mode and for one hour of integration time.

measures circular polarization, so only when the magnetic field is pointing at the sensor will there be a strong signal. For nadir staring mode, this limits us to the stronger magnetic field areas, which covers a relatively small area. So to achieve the error here, very many subsequent orbits must observe the area. For limb scanning mode, combining the measurements above a single tangent profile from different azimuth angles will improve the precision even more than shown in our figures. Finally, the molecular oxygen profile in our simulations consists of almost $50\%$ less molecular oxygen than has previously been measured because these are produced by the circulation model. A $40\%$ increase in molecular oxygen in our simulations means that the estimated error is reduced by about a third of the error estimates given here.

- Temperature: In limb scanning mode, the temperature error estimated for a single profile is about $2\,\mathrm{K}$ to $2.5\,\mathrm{K}$ with good measurement response from $10\,\mathrm{km}$ to $40\,\mathrm{km}$. In nadir staring mode, the temperature error estimation is about $1.5\,\mathrm{K}$ with good measurement response from ground to $50\,\mathrm{km}$.

- Wind speed: The wind speed error is estimated from $20\,\mathrm{m\,s^{-1}}$ to $25\,\mathrm{m\,s^{-1}}$ depending on atmospheric wetness, for good measurement response at an altitude range approximately from $20\,\mathrm{km}$ to $60\,\mathrm{km}$. In nadir staring mode, wind speed errors are about $20\,\mathrm{m\,s^{-1}}$ with good measurement response from $10\,\mathrm{km}$ to $60\,\mathrm{km}$. The vertical wind speed will not be this strong along the path of observations in nadir staring mode, so attempting to retrieve the wind parameter with our instrument in nadir observations will not be useful.

**3.3 On Vertical Resolution v. Precision**

An estimation of the vertical resolution versus precision can be had by letting the correlation distance increase to large distances. In Figure 8 we let the correlation distance vary from a diagonal a priori matrix to $1000\,\mathrm{km}$ for both nadir staring and limb scanning simulations. The figure also shows the degrees of freedom of of the averaging kernel at these correlation distances, mostly to indicate that what we are trading for increased precision is lower degrees of freedom. The lower correlation distance has already been discussed above so below we only discuss the upper range. The upper range represents a scenario where we assume that the state of the atmosphere at one level is highly correlated to that at every other level. This is a somewhat crude way to reduce the degrees of freedom of the problem and thereby estimate the error at a lower vertical resolution, though it should be sufficient for our purposes. For molecular oxygen, this highly correlated state reduces the estimated errors in the nadir staring mode to about 10 ppmv, and in the limb scanning mode to about 15 ppmv. For gaseous water, the highly correlated state reduces the error estimation to 0.1 ppmv for both nadir staring and limb scanning mode. For hydrogen peroxide, the estimated errors are reduced in the highly correlated state to about 1 ppbv in nadir staring and to 0.5 ppbv in limb scanning mode. The limb scanning mode retains more than one degree of freedom. For ozone, limb scanning mode errors remain at about 2 ppbv in the highly correlated state as its degrees of freedom are low to begin with for the uncorrelated a priori covariance matrix. The magnetic field in the highly correlated state has limb scanning mode estimated errors at $1\,\mathrm{\mu T}$ and in nadir staring at $0.5\,\mathrm{\mu T}$. The temperature error estimation in the highly correlated case retains degrees of freedom in both nadir staring and in limb scanning mode, meaning the retrieval system does not allow the trade that we are after. The errors are expected to be about $1\,\mathrm{K}$. The wind speed also retains degrees of freedom in both viewing geometries. The errors are expected to be around $10\,\mathrm{m\,s^{-1}}$. Since the magnetic field is highly correlated with distance, these numbers might better represent the precision at which we can retrieve the magnetic field strength than the errors estimated in the previous subsection. For the other parameters, the correlation distances are too long so the number represents a limit at poor vertical resolution for a single profile measurement.

**4 Conclusions**

We present a feasibility study for a sub-millimeter instrument on a small Mars mission currently under construction. The instrument will be able to measure several atmospheric species crucial for understanding Martian molecular oxygen, gaseous

[Figure]

**Figure 8.** Correlation distance versus estimated errors and degrees of freedom (DOF). Panel (a) is for molecular oxygen. Panel (b) is for gaseous water. Panel (c) is for hydrogen peroxide. Panel (d) is for ozone. Panel (e) is for the magnetic field strength. Panel (f) is for the temperature. Panel (g) is for the wind speed. Legends indicate if the line belongs to the right y-axis or left y-axis, as well as the observation strategy as used for Figures 5 or 7. The estimated errors are the mean over the range of $\max(\mathbf{r}) > 0.8$, so the ozone-panel lacks a line for nadir staring mode since it is never sensitive at the a priori standard deviations presented in the subsection on error analysis.

water, hydrogen peroxide, and ozone at low errors. In addition to these, the instrument will be able to limit the crustal magnetic field, and get the meteorological parameters of temperature and wind speed. The expected measurement errors before spatial

and temporal averaging are 15 ppmv to 25 ppmv for the molecular oxygen mixing ratio, 0.2 ppmv for the gaseous water mixing ratio, 2 ppbv for the hydrogen peroxide mixing ratio, 2 ppbv for the ozone mixing ratio, $1.5\,\mu\text{T}$ to $2.5\,\mu\text{T}$ for the magnetic field strength, $1.5\,\text{K}$ to $2.5\,\text{K}$ for the temperature profile, and $20\,\text{m}\,\text{s}^{-1}$ to $25\,\text{m}\,\text{s}^{-1}$ for the horizontal wind speed. These error ranges are well within the range to push the boundary of Mars knowledge.

---

## Author Comment (AC2) · 23 Oct 2018

**1  General changes**

The changes to the paper between this and previous version are substantial. The major changes are as listed

- The error analysis has been redone completely based on overarching comments from both reviewers. This has been done mainly to be able to discuss what both reviewers mentioned as important, namely the trade off between precision and resolution in an added subsection at the end of section 3. This discussion is not exhaustive but we think it addresses the reviewers' concerns about the the

aforementioned trade off.

- The forward simulations have been changed by using available GCM model atmospheres and updating the Zeeman effect spectroscopy. The former was done to better represent cases with a changed correlation distance (or scale length) in the a priori covariance matrix required to discuss mentioned trade off. The latter is a more recent development where the main author found a small (a few percent) error in the Zeeman splitting for the used molecular oxygen line.

Other smaller changes have also been incorporated in the current version.

Below in red are the reviewers' comments one-by-one as copy-pasted from the original response files provided by Copernicus. The one alteration that the main author has done to these comments is to add appropriate LATEX-coding for references to variables in equations.

Before going over the detailed response, we wish to thank the reviewers because their suggestions have improved this work substantially.

Attached is the updated paper.

**2  Respons to reviewer 2**

*The manuscript is about using sub-millimeter radiometry to measure Martian atmosphere. In my opinion this idea is a solid one and combining the sub-millimeter instrument with a micro satellite platform follows the current trend.*

*However, I find some problems with this feasibility study. I do see that the main emphasis of the manuscript is on the radiometer instrument and the measurements made with it. I would still like to see the space technology details discussed more.*

There is a paper under preparation about the platform. We will therefore not discuss the platform in much more details in this work.

*For example, the authors state that putting the satellite into Martian orbit has never been attempted before using atmospheric drag. The authors just assume that this can be done and assume a couple of potential orbits. But doesn't this mean that using micro satellite platform makes the mission harder to accomplish? Why not just put the instrument to a bigger platform and put it into orbit by more conventional means?*

We wish to be clear, we state that using atmospheric drag as the only breaking mechanism has never been done before. The paper under preparation will discuss these points in more detail.

The reviewer is very much correct that our instrument can be placed on any other type of platform that wish to accept it as part of their payload, and that the small platform makes the mission much harder than if we could use a larger platform. To answer the second question a bit bluntly, we working on using the small scale approach because flying to Mars is expensive, we know of no one with an open call that would accept a sub-millimeter sensor on a Mars mission, and with a small platform we should be able to get this done ourselves.

None of these comments seem to fit in the paper, so the text has not been updated.

*I also think that the description of the radiometer instrument is lacking. I did read the Kasai et al. 2012 paper and they describe the FIRE radiometer in detail. In what way the FIRE-mini differs from the FIRE instrument? Is it only size as the authors mention or the usage of two channels with different circular polarization? How about weight and power budget? Kasai et al. 2012 gives two options of FIRE radiometer; Limited science: 5 kg, 10 W & Full science: 16 kg, 40 W. If the micro satellite platform has max. weight of 100 kg then fitting in even the full science option of FIRE might do the job. So what is the advantage of FIRE-mini over FIRE?*
The mass and power budget is smaller.

We update the start of subsection 2.2 to exemplify one limitation the platform has incurred on us that did not concern FIRE.

We have also changed the name working name internally to TEREX as an official name for the mission.

*Solar power is available less in the Martian orbit than in Earth orbit. Is the 40W power feasible using solar panels? In my opinion the small satellite size could be a limitation in this case. How about the attitude control? A probable choice would be magnetic control which also uses electric power. So what is the actual advantage of using micro satellite platform? Is it the reduced costs?*

Indeed, the power budget is very limited. The exact number we have is still a work in progress. The paper discussing the platform will go over these details. It is beyond this work.

*I think the Reviewer #1 has good comments about the trade offs in accuracy, resolution and precision. The Reviewer #1 also points out that making the radiometer more sensitive reduces the errors in parameter retrieval. This could be difficult since the receiver is heterodyne receiver (because of sub-millimeter wavelengths) or at least depends much on the calibration of the instrument. I suppose the calibration of the FIRE-mini will be done using 2.7K background microwave radiation as with FIRE? Even though the orbit selection might no have a big effect on those the attitude control probably will have. Magnetic attitude control, however, is not that precise.*

Yes, space is the background. The limb scanning mode we describe will not be as accurate as in figure 4, with perfect staring at each tangent altitude. Making it accurate either before making the measurements or as a reconstruction a posteriori is going to be a major challenge, and the exact limitation of this is going to have to be quantified once the measurements are available. Figure 4 and later figures should still fairly accurately represent the error estimations if we can somehow reliably reconstruct pointing afterwards.

*The authors describe the forward model in chapter 2.3 and then I suppose chapter 2.4 describes how the inversion of forward model is done. I think the description is more on the conceptual level and it is not that easy to see what is actually done in the paper.*

Hopefully the update fixes most concerns about the inversions. We cite four papers describing some parts of the forward model, and one additional paper describing the spectroscopic parameters. The forward model is also available to read via provided hyper-links if even more in-depth details are necessary.

Please also note the supplement to this comment:
https://www.geosci-instrum-method-data-syst-discuss.net/gi-2017-50/gi-2017-50-AC2-supplement.pdf